# Treatment Sequencing in Metastatic HR+/HER2− Breast Cancer: A Delphi Consensus

**DOI:** 10.3390/cancers17091412

**Published:** 2025-04-23

**Authors:** Lazar Popović, Simona Borštnar, Ivana Božović-Spasojević, Ana Cvetanović, Natalija Dedić Plavetić, Radka Kaneva, Assia Konsoulova, Erika Matos, Snježana Tomić, Eduard Vrdoljak

**Affiliations:** 1Oncology Institute of Vojvodina, Faculty of Medicine, University of Novi Sad, 21000 Novi Sad, Serbia; 2Institute of Oncology Ljubljana, 1000 Ljubljana, Slovenia; sborstnar@onko-i.si (S.B.); ematos@onko-i.si (E.M.); 3Institute for Oncology and Radiology of Serbia, Medical Faculty, University of Belgrade, 11000 Belgrade, Serbia; ivanabs@ncrc.ac.rs; 4University Clinical Centre Niš, Medical Faculty of Niš, 18000 Niš, Serbia; ana.cvetanovic@medfak.ni.ac.rs; 5University Clinical Hospital Center Zagreb, School of Medicine, University of Zagreb, 10000 Zagreb, Croatia; 6Molecular Medicine Center, Medical University of Sofia, 1431 Sofia, Bulgaria; kaneva@mmcbg.org; 7University Cancer Hospital Prof. Ivan Chernozemski, 1756 Sofia, Bulgaria; akonsoulova@sbaloncology.bg; 8Department of Preclinical and Clinical Disciplines, Faculty of Social Health and Healthcare, University Prof. A. Zlatarov, 8000 Burgas, Bulgaria; 9Clinical Hospital Center Split, University of Split School of Medicine, 21000 Split, Croatia; stomic@mefst.hr (S.T.); eduard.vrdoljak@mefst.hr (E.V.)

**Keywords:** HR+/HER2− metastatic breast cancer, second-line therapy, targeted therapy, treatment choice, Delphi consensus

## Abstract

Metastatic breast cancer (mBC) carries a huge burden for patients and healthcare systems globally. Optimal treatment is of paramount importance to streamline the treatment journey. In HR+/HER2− mBC, at disease progression on first-line therapy, the choice of next treatment lines should be guided not only by the presence of specific targetable mutations, but also by evidence of efficacy and safety from clinical trials and access to genetic testing and medications. The aim of the Delphi process is to gain consensus (at least 70% agreement) on the perspective of treatment strategies from experts in the field. The outcome of Delphi discussions is an algorithm for the second line in HR+/HER2− mBC. Clinicians may find it useful in their current practice and use it as a basis for the treatment individualization strategy, which must remain the core principle of our actions.

## 1. Introduction

Breast cancer (BC) is one of the most common malignancies in women, representing approximately 30% of all cancers [1,2]. Overall, 70–85% of the tumors are characterized by the presence of hormonal receptors (HR+) and absence of HER2 overexpression or HER2/neu gene amplification (HER2− and HER2-low) [3,4].

At the global level, the number of new cases will increase by 38% in 2050, and the number of deaths due to breast cancer by 68%, a projection that reinforces the need for continuous research in oncology and early access to treatment [5].

In recent years, nationwide screening programs have improved the detection of BC, allowing initiation of systemic therapies earlier, in early or metastatic settings, resulting in improved outcomes in both [6]. Progression may occur in months or years from initial diagnosis and treatment. Metastatic disease is generally considered incurable and is the driver of mortality in BC.

Based on recently published data from robust phase III trials, the addition of inhibitors of the cyclin-dependent kinase 4/6 (CDK4/6i) to endocrine therapy (ET) is the current standard of the first-line (1L) treatment in ER+ HER2− metastatic BC (mBC) [7,8,9,10]. Compared with ET alone, the combination improved progression-free survival (PFS) and proved good tolerability without any related deterioration of the quality of life [11,12,13,14]. Moreover, the combination of ET and a CDK4/6i was compared with dual chemotherapy in patients with aggressive tumor characteristics, and the results showed similar efficacy with a better safety profile [15,16]. Currently, the European Society of Medical Oncology (ESMO) guidelines recommend chemotherapy as first-line systemic therapy only in patients with imminent organ failure, while endocrine monotherapy is reserved for a small group of patients with comorbidities or a poor performance status and for tumors likely to respond to chemotherapy better (e.g., low estrogen receptor status, high Ki67) [8].

First-line treatment is almost inevitably followed by disease progression. Previous response and duration of response to adjuvant and first-line therapy, comorbidities, and the patient’s preferences, as well as the potential toxicity risks, are important factors in planning the management in the advanced stage. Despite the availability of several biomarker-driven treatment options and updated algorithms [7,8,17,18], the optimal sequence after a CDK4/6i is still not clear [18,19], especially in patients with concomitant genomic alterations. The uptake of guidelines is often questionable, especially in underserved medical systems. Consequently, the development of guidelines that are clinically and pharmacoeconomically widely applicable is of special importance.

To further explore this gap, a Delphi technique has been adopted to provide evidence-based recommendations on this important medical challenge, considering the wide use of such a process in healthcare and specifically in oncology [20,21,22,23]. The manuscript summarizes the consensus approach and the current perspectives on the second-line treatment in HR+/HER2− mBC.

## 2. Methods

The panel for the diagnosis, management, and monitoring of mBC patients from four countries in the Balkan region included ten experts. One of the members was the scientific consultant who reviewed the available literature and identified several topics of interest related to the choice of treatment in HR+/HER2− mBC. The Delphi survey included statements grouped into four sections as follows: (1) biomarker testing; (2) selection of 2L treatment at progression of disease on 1L endocrine therapy (ET) + CDK4/6i at ≥6 months after initiation of ET for mBC whilst on ET; (3) selection of 2L treatment at PD on 1L ET + CDK4/6i, at <6 months after initiation of ET for mBC, whilst on ET; (4) selection of post-2L treatment options (progressive disease after minimum 2 lines of therapy for mBC). The questionnaire has been developed with Google Forms and sent via email for anonymous voting to participants. They expressed agreement or disagreement on a 5-item Likert scale (“completely agree”, “agree”, “neutral”, “disagree”, and “completely disagree”). Consensus was defined as at least 70% agreement (including “completely agree” and “agree” answers). The 70% threshold is a good indicator for recommending a certain strategy in clinical practice. This level was also used in other similar projects in oncology [20,22].

After voting, the scientific consultant compiled and prepared the results from the first round and served as a facilitator during the Delphi process. A virtual meeting was held on 4 July 2024 to discuss the results, provide feedback, and decide on statements where consensus has not been reached. Statements that were newly created or modified were sent for voting in a second round to the same group of participants. A prespecified level of agreement has been reached, and another round of voting and discussion was not required.

The recommendations, supported by this Delphi process, reflect the opinion of invited experts and are intended to guide medical oncologists in treatment choices starting with a second line in an HR+/HER2− metastatic setting. While they are based on published evidence and reflect general strategies, the experts are aware that clinical practice may be different and patient management will be the result of clinical judgment, patient preferences, as well as access to testing and novel therapies [24,25,26].

## 3. Results

The Delphi process consisted of two rounds of voting and one virtual meeting. Out of 39 statements, consensus was achieved in 30 of them following the first round of voting. After providing feedback during the virtual meeting, the experts decided to remove four statements, refine ten, and include three new statements. The participants voted on four statements in the second round. The final list consisted of 38 items for which at least 70% agreement or disagreement was achieved in 37 statements. All statements refer to the management of HR+/HER2− mBC after disease progression on the 1L ET plus CDK4/6 inhibitor, before 2L and beyond. For each statement, the level of agreement, following each round of voting, is presented in Table 1.

## 4. Discussions

### 4.1. NGS Is the Preferred Testing Method for Accurate Molecular Characterization

The diagnosis of BC still relies on immunohistochemistry (IHC) techniques to determine the histology and receptor status. The identification of the hormonal receptor presence was the first biomarker test and formed the basis for building the treatment strategy. Determination of HER2 expression and its therapeutic and prognostic roles significantly changed the breast medical oncology field. An assessment of HER2 status by IHC, combined with in situ hybridization (ISH), led to differentiation of additional categories (0, 1+, 2+, and 3+) and introduced the concept of HER2-low and -ultralow expression; however, this has not been completely integrated into routine clinical practice [24]. At disease progression, reassessment of the tumor immunophenotype through biopsy is currently recommended [8,9].

Breast cancer is marked by histological and molecular heterogeneity. Under the selective pressure of the tumoral microenvironment and treatment, the genomic profile changes by acquiring additional mutations that are important for cell survival. Metastatic lesions often present substantially different genomic alterations from the primary tumor [27], including changes in HER2 expression and hormonal receptor status between primary tumors and the related distant metastases, and between different metastatic sites [28,29]. In addition to hormonal receptors and HER2 expression, guidelines recommend testing additional biomarkers to inform the treatment decision from the second line [8,9]. In HR+/HER2− mBC, gene expression-based signatures, Sanger sequencing, and next-generation sequencing (NGS) are employed to detect targetable genetic mutations, prompting the initiation of targeted therapies and extending the time to chemotherapy. Of all, NGS combines the gains of advanced chemistry and digital technologies [30] to analyze an immense quantity of DNA and RNA sequences or even the whole genome in a relatively short time and acceptable cost compared to other sequencing techniques. It is the method of choice, currently recommended by ESMO guidelines, to detect multiple clinically relevant mutations present in HR+/HER2− mBC at disease progression. The 2024 ESMO Scale for Clinical Actionability of Molecular Targets (ESCAT) indicates NGS as standard for the identification of *ESR1*, *PIK3CA*, and germline *BRCA1/2* mutations (level IA). Moreover, *PTEN* and *AKT1* alterations were provided with a level I/II score in this patient population [31]. The current version of the National Comprehensive Cancer Network^®^ guidelines indicates NGS for the identification of these mutations, along with alternative methods where applicable [10].

During the initial voting, an agreement has not been reached on NGS testing to guide second-line therapy (60% agreement). The discussion during the virtual meeting helped to clarify the topic. Although NGS is considered the method of choice, based on the arguments presented above, other techniques may be currently more frequently used in clinical practice to determine mutational status, such as PCR for *PIK3CA* mutation, digital droplet PCR for *ESR1* mutation, IHC and ISH, which are still recommended for defining HER2 protein overexpression or gene amplification. The panel decided to replace the first remark with two different statements, and 100% agreement was reached for both (Table 1). Nevertheless, the choice of therapy at disease progression is based on the presence of targetable mutations, previous exposure to, and duration of the response to ET (primary or secondary resistance) (Figure 1).

### 4.2. Treatment Choice

The combination of ET with a CDK4/6i is the standard of care in the first-line setting in patients with HR+/HER2− mBC [7,8]. The use of chemotherapy at mBC diagnosis is reserved for cases with imminent organ failure or life-threatening visceral disease. After a median progression-free survival (mPFS) of approximately 25 months [11,12,14], disease progression occurs by acquired resistance to ET and/or CDK4/6i.

The treatment choice upon disease progression should consider the presence of mutations, the previous exposure and response to CDK4/6i and endocrine partner, and evidence from clinical trials, as well as patient status, comorbidities, and preferences. The decision may be easier in the presence of specific targeted mutations (Figure 1) and available therapies; however, many other, more common situations are not so straightforward in pointing towards a certain medicine or combination approach. Although ESMO and NCCN guidelines have been recently updated [8,10] and the decision tree is important to the overall strategy, the Delphi process helped with more direct guidance regarding clinical practice.

#### 4.2.1. *ESR1* Mutation

*ESR1* is a dynamic mutation linked with dominantly acquired endocrine resistance. Most experts recommend testing at the time of disease progression since it is very rarely detected before any treatment initiation (up to 5%), while the prevalence is up to 50% in patients with mBC previously exposed to an aromatase inhibitor (AI) [33,34].

Important to note, the monitoring of *ESR1* using liquid biopsy in the PADA-1 study revealed that the method may be successfully applied to identify treatment resistance [35]. Switching to a different therapy as soon as *ESR1* mutation is detected and before the radiologic progression results in survival benefits. More results are expected from the ongoing SERENA-6 trial investigating the efficacy and safety of a treatment switch from AI to a next-generation oral selective estrogen receptor degrader (SERD) camizestrant before clinical disease progression with the first-line therapy [36]. The routine use of circulating tumor DNA (ctDNA) requires standardization of the workflow and molecular reporting [37] for reliable results. The method is indicated as preferred in the current NCCN Guidelines [10] to identify *ESR1* mutation.

In terms of treatment choice, the open-label, phase III, EMERALD trial showed the benefit of elacestrant on mPFS, compared with investigators’ choice of fulvestrant or an AI (exemestane, letrozole, or letrozole) in HR+/HER2− mBC patients with *ESR1* mutation and previous exposure to CDK4/6i (mostly palbociclib) [38]. While the initial improvement was small (3.8 months vs. 1.9 months), the Kaplan–Meier survival curves showed significant divergence between the groups as early as 6 months, an indicator of the limited efficacy of ET monotherapy at disease progression [39]. In patients with *ESR1* mutation, prolonged exposure to ET + CDK4/6i of more than 12 months in a metastatic setting resulted in longer mPFS with elacestrant compared to standard of care (8.6 months vs. 1.9–2.1 months) [34,39] (Table 2).

A recent comprehensive review of results from clinical trials presents the evidence on efficacy and safety of oral SERDs compared with standard treatment in HR+/HER2− mBC patients with *ESR1* mutation [40]. New SERDs have the potential to become the ET backbone with targeted therapies [41].

**Table 2 cancers-17-01412-t002:** Summary of clinical trials with agents used in 2L+ HR+/HER2− mBC.

**Study name/phase**	**MAINTAIN/II** [42]	**PACE/II** [43]	**postMonarch/III** [44]	**EMERALD/III** [38,39]
	NCT02632045	NCT03147287	NCT05169567	NCT03778931
Study population	Ribo + ET vs. pbo + ET	F + pbo vs. fulvestrant + palbo ± avelumab	F + pbo vs. F + abema	Elacestrant vs. AI or F
Number of patients	119	220	368	477
Prior lines of therapy in a metastatic setting	≤1 ChT	≤1 ChT	ET + CDK4/6i	≤1 ChT
Prior exposure to CDK4/6i in mBC	86.5% to palbo 11.7% to ribo	100%	100% 59% palbo, 33% ribo	100%
mPFS	5.3 vs. 2.8 months; HR = 0.56; 95% CI 0.39–0.85; *p* = 0.006	F: 4.8 months; F+ palbo: 4.6 months (HR, 1.11 [90% CI, 0.79 to 1.55]; *p* = 0.62). F + palbo + avelumab: 8.1 months (HR vs. F, 0.75 [90% CI, 0.50 to 1.12]; *p* = 0.23)	Interim analysis: HR = 0.66; 95% CI 0.48–0.91; *p* = 0.01 Final analysis: HR = 0.73; 95% CI 0.57–0.95	*ESR1*m: 3.8 months vs. 1.9 months; HR = 0.55 (95% CI 0.39–0.77; *p* = 0.0005) ≥12 months prior exposure to CDK4/6i: 8.6 months vs. 1.9 months (HR = 0.410; 95% CI 0.262–0.634)
mOS	NR	NR	OS data are immature	ESR1m: HR = 0.59 (95% CI, 0.36–0.96; *p* = 0.03, nonsignificant)
Other relevant results	F was the ET backbone in 83.2% of participants; Neutropenia rates were higher in the ribo arm			The most common AEs: GI events, fatigability, and arthralgia
**Study name/phase**	**SOLAR-1/III** [45,46]	**BYLieve/II** [47]	**CAPITello-291/III** [48,49]	**OlympiAD/III** [50,51,52]
	NCT02437318	NCT03056755	NCT04305496	NCT02000622
Study population	Alpe + F vs. pbo + F	Alpe + F or Let	Capi + F vs. pbo + F	Olaparib vs. ChT
Number of patients	572	127	708	302, g*BRCA1/2*m
Prior lines of therapy in a metastatic setting			≤3	≤2 ChT
Prior exposure to CDK4/6i in mBC	~5%	100% in 2 cohorts	69%	No
mPFS	*PIK3CA*m: 11.0 months vs. 5.7 months, HR = 0.65; 95% CI 0.50–0.85, *p* < 0.001	Pre-treated with CDK4/6i + AI: 8.0 (5.6 to 8.6) months CDK4/6i + F: 5.6 (3.7 to 7.1) months	*PI3K/AKT/PTEN*: 7.3 vs. 3.1 months (HR = 0.50, 95% CI 0.38–0.65; *p* < 0.001)	7.0 vs. 4.2 months (HR = 0.58; 95% CI 0.43–0.80, *p* < 0.001)
mOS	*PIK3CA*m: 39.3 vs. 31.4 months, HR = 0.86 (95% CI, 0.64–1.15; *p* = 0.15)	Pre-treated with CDK4/6i + AI: 27.3 (21.3–32.7) months CDK4/6i + F: 29.0 (24.5–34.8) months	NR	At 64% data maturity: 19.3 vs. 17.1 months (HR 0.90, 95% CI 0.66–1.23; *p* = 0.513) At 76.8% data maturity: 19.3 vs. 17.1 months (HR = 0.89, 95% CI 0.67–1.18)
Other relevant results	The most common AEs of grade 3 or 4: hyperglycemia (36.6% vs. 0.7%) and rash (9.9% vs. 0.3%)	The most common AEs: GI events, hyperglycemia, rash, fatigability. Hyperglycemia was the most common grade ≥ 3 AE.	The most common AEs of grade 3: rash (12.1% vs. 0.3%) and diarrhea (9.3% vs. 0.3%). Capi + F delayed time to deterioration of GHS/QOL and maintained other dimensions of HRQOL (except symptoms of diarrhea), similarly to F	Anemia, nausea, vomiting, fatigue, headache, and cough occurred more frequently in the olaparib group than in the standard-therapy group; AEs events during olaparib treatment were generally low grade and manageable by supportive treatment or dose modification. Long-term exposure to olaparib was generally well tolerated, with no evidence of cumulative toxicity and no new safety signals
**Study name/phase**	**EMBRACA/III** [53,54]	**DESTINY-Breast04/III** [55,56]	**DESTINY-Breast 06** [32]	**TROPICS-02/III** [57,58]
	NCT01945775	NCT03734029	NCT04494425	NCT03901339
Study population	Talazoparib vs. ChT	T-Dxd vs. ChT	T-DXd vs. ChT	SG vs. ChT
Number of patients	431, g*BRCA1/2*m	494, HR+/HER2-low	866 (713 with HER2-low)	543, HR+/HER2-low
Prior lines of therapy in a metastatic setting		Median 3 (range 1–9)	ET, no chemotherapy	Median 7 (range 3–17)
Prior exposure to CDK4/6i in mBC	No	71%	88.6% in the T-DXd group 89.3% in the ChT group	~100%
mPFS	8.6 vs. 5.6 months (HR = 0.54; 95% CI 0.41–0.71, *p* < 0.001)	10.1 vs. 5.4 months (HR = 0.51, 95% CI 0.40–0.64; *p* < 0.001)	13.2 months vs. 8.1 months; HR = 0.62; 95% CI 0.52–0.75; *p* < 0.001	5.5 vs. 4.0 months (HR = 0.66, 95% CI 0.53–0.83, *p* = 0.0003)
mOS	At 57% data maturity: HR = 0.76; 95% CI 0.55–1.06; *p* = 0.11 Final mOS: 19.3 vs. 19.5 months (HR = 0.848; 95% CI 0.670–1.073; *p* = 0.17)	23.9 vs. 17.6 months (HR = 0.69, 95% CI 0.55–0.87)	OS data are immature	14.4 vs. 11.2 months (HR = 0.79, 95% CI 0.65–0.96, *p* = 0.02)
Other relevant results	The most common AEs were hematological; PROs favored talazoparib, with significant overall improvement and delays in time to clinically meaningful deterioration.	The most common TEAEs were GI and hematological; ILD/pneumonitis: 12% vs. 1%; LV dysfunction: 13% vs. 6%	Similar incidence of AEs with T-DXd (98.8%) and ChT (95.2%). The most common drug-related AEs: nausea, fatigue, and alopecia (T-DXd), and fatigue, palmar–plantar erythrodysesthesia syndrome, and neutropenia (ChT)	The most vs. 54%common TEAEs were GI and hematological; nausea: 55% vs. 31%; neutropenia: 70%

Abbreviations: 2L, second line; abema, abemaciclib; AE, adverse event; AI, aromatase inhibitor; alpe, alpelisib; capi, capivasertib; CDK4/6i, cyclin-dependent kinase 4/6 inhibitor; CI, confidence interval; ChT, chemotherapy; F, fulvestrant; g*BRCA1/2*m, germline *BCRA1* or *BRCA2* mutation; GHS/QOL, health-related quality of life; GI, gastrointestinal; HR, hazard ratio; HR+, positive hormone receptors; ILD, interstitial lung disease; Let, letrozole; LV, left ventricle; mBC, metastatic breast cancer; mPFS, median progression free survival; mOS, median overall survival; NR, not reported; pbo, placebo; palbo, palbociclib; PRO, patient-reported outcome; ribo, ribociclib; SG, sagituzumab govitecan; T-Dxd, trastuzumab deruxtecan; TEAE, treatment emergent adverse event.

#### 4.2.2. *PIK3CA*/*AKT*/*PTEN* Alteration

Amplification or activating mutations occurring on the *PI3K* signaling pathway are frequent events in BC, with a reported prevalence of up to 50% of the HR+ cases [59,60].

Recent reports indicate that *PIK3CA* mutations are present in 30–40% of the ER+/HER2− subgroup [60,61,62,63], with most patients being previously exposed to CDK4/6i. The prevalence of *PIK3CA* mutations is either maintained [61] or increased [60] with the lines of therapy. The *PIK3CA* mutation is stable; therefore, the testing performed on tissue from the primary tumor would lead to similar results with biopsies from the metastatic lesions. More research is needed to clarify the relationship between *PIK3CA* mutation status and primary endocrine resistance, since current published reports show contradictory results [62,64], with implications for testing before 1L or after disease progression.

While alpelisib plus fulvestrant represents a valid option in HR+/HER2− mBC [7,8,10], the experts preferred the combination of capivasertib plus fulvestrant (Figure 1). In the absence of comparative trials, the choice is based on similar mPFS results observed in CAPItello-291 and BYLieve trials, since the SOLAR-1 phase III trial included only a very low number of patients previously exposed to a CDK4/6i; thus, results could not be extrapolated to the contemporary strategy dominated by CDK4/6i use [45,46,47,48]. Moreover, the OS improvement with alpelisib is only numerically and not statistically significant. The tolerability profile had a decisive influence on the agreement [9]. The difference between the two drugs is related to increased rates of grade 3 hyperglycemia with alpelisib (36.6%) compared to capivasertib (2.3%), with higher discontinuation rates in alpelisib trials (Table 2). Hyperglycemia is an early and frequent adverse event during alpelisib treatment [46]. The adoption of prophylaxis with metformin may improve alpelisib tolerability [65,66,67] while increasing the rates of gastrointestinal side effects, mainly diarrhea, adding more burden to the disease management. Nevertheless, alpelisib is indicated only in patients with *PIK3CA*-activating mutations but not *AKT1*-activating mutations or *PTEN* inactivation.

The introduction of inavolisib in the first-line therapy of *PIK3CA* mutated HR+/HER2− mBC [10] is about to make the treatment strategy process even more complex from the diagnosis of advanced disease; thus, a personalized treatment approach is of paramount importance [68].

The strategy, combining a PI3Kα inhibitor with a CDK4/6i and ET, was tested in 1L HR+/HER2− advanced BC on patients with primary endocrine resistance (disease progression during or <12 months from adjuvant ET) [68]. After almost one year of follow-up, the mPFS was 15.0 months with inavolisib plus palbociclib and fulvestrant, compared to 7.3 months in the group with palbociclib plus fulvestrant (HR = 0.43, 95% CI 0.32–0.59, *p* < 0.0001) [68]. Although the study included only patients without diabetes (fasting plasma glucose < 126 mg/dL or HbA1c 6.0%), the hyperglycemia levels reported in the triple therapy group were high (58.6%) [68,69]. A similar percentage has been observed for stomatitis/mucosal inflammation (51.2%) [68]. The impact of such a strategy for the 2L choice and beyond is to be evaluated by future studies.

#### 4.2.3. Germline *BRCA1/2* Mutation

NGS helps with the identification of *BRCA1/2* pathogenic or likely pathogenic variants, both germline and somatic. Robust clinical trials [50,53] demonstrated the efficacy and safety of the poly (ADP-ribose) polymerase inhibitors (PARPis) in patients with HER2-negative advanced breast cancer and pathogenic germline *BRCA1/2* mutations (g*BRCA*m). Phase II trials are investigating the role of PARPis in patients with breast cancer and somatic *BRCA* mutations [70,71].

By refining the original statement 2 of the Delphi questionnaire, the panel stressed the importance of available results of germline mutation testing at disease progression as a key treatment decision factor. Both olaparib and talazoparib have demonstrated improvement in mPFS vs. standard chemotherapy in patients with metastatic breast cancer and a pathogenic g*BRCA*m and are indicated in this setting [10] (Table 2); however, no statistical OS improvement with talazoparib or olaparib was found [50,51,53,54].

Almost 50% of patients included in the OlympiAD [50] and EMBRACA [53] trials had HR+/HER2− tumors, and the presence of g*BRCA*m seems to be associated with a poor response to frontline ET plus CDK4/6i [72]. Although no data are available from prospective clinical trials using a PARPi after a CDK4/6i, the experts recommend this approach if g*BRCA*1/2 carriers are present to delay the time to chemotherapy initiation.

#### 4.2.4. Concomitant Mutations

Systematic screening programs helped to diagnose more people with breast cancer in the early stages, treat them accordingly, and reduce overall mortality [73]. A longer time with disease and a longer treatment duration increase the odds of accumulating concomitant mutations [74]. The transcriptional landscape in elderly patients with breast cancer is dominated by concomitant somatic mutations, such as *ESR1* and *PIK3CA* [75]. Therefore, the presence of multiple targetable alterations and lack of dedicated sequencing trials could make the treatment decision difficult starting with the second line.

In approximately half of patients with ER+/HER2− mBC, at least one genomic alteration (*ESR1*, *PIK3CA*, *PTEN*, or *AKT*) may be identified by the time of the initiation of the first line of therapy. The prevalence is increasing in later lines and longer metastatic disease due to *ESR1* mutation acquisition [74,76]. The co-occurrence of *ESR1* and a *PI3K/AKT* mutation may be found in up to 8% of patients, initiating second-line therapy in ER+/HER2− mBC [60]. The concomitant presence of *PIK3CA* and *PTEN/AKT1* mutations was observed in low percentages, irrespective of the time of assessment [60,77]. *AKT* and *PTEN* mutations are mutually exclusive.

Capivasertib plus fulvestrant, as well as elacestrant, would qualify as preferred options for the expert group [7,10]. This is the only statement where a consensus level for the choice of one strategy over the other has not been reached (Table 1). The therapeutic approach should be based on the duration of the response to the previous line with ET plus CDK4/6i (≥12 months vs. <12 months) (Figure 1) [34].

#### 4.2.5. HR+/HER2− mBC and No Actionable Mutations

Switching the CDK4/6i and/or ET may be discussed, since improvement in mPFS has been observed with ribociclib or abemaciclib plus ET [42,44] in patients with HR+/HER2− mBC previously exposed to palbociclib. No positive outcomes have been observed when continuing the use of palbociclib upon progression [43,78] (Table 2); therefore, this strategy is not recommended by the experts. The maintenance of a CDK4/6i after progression is placed under question by an exploratory ctDNA analysis suggesting that *ESR1* or *PIK3CA* mutation would result in no benefit regarding the addition of ribociclib to ET [9], which makes genetic testing more accurate and the treatment decision more important. A reassessment of HER2 expression at disease progression may help to identify a group of patients that may benefit from antibody–drug conjugates in endocrine-resistant settings. Remarkable improvements in mPFS have been achieved with sacituzumab govitecan (SG) (TROPiCS-02 trial, in a heavily pretreated population with a median of three lines of prior chemotherapy) and trastuzumab deruxtecan (T-DXd) (DESTINY-Breast04, after chemotherapy and DESTINY-Breast06 in endocrine-insensitive patients before chemotherapy) [32,55,57]. Current ESMO guidelines [7,8] recommend SG for patients with HR +/HER2-0 mBC after at least two lines of ChT and T-DXd for patients with HR+/HER2-low mBC after at least one line of ChT. The DESTINY-Breast06 trial showed significant improvement in progression-free survival with T-DXd compared to standard chemotherapy in patients with HR+/HER2-low and HER2-ultralow mBC [32]. In this patient population (HER2-low and -ultralow), T-DXd may become the preferred choice after at least one line of ET, before chemotherapy, with SG to be considered in a later line.

## 5. Conclusions

The treatment landscape in HR+/HER2− mBC continues to expand with novel agents, pushing the survival barriers further. ET plus CDK4/6i is the current standard frontline for most patients except for imminent organ failure due to malignant disease, including fit elderly patients and high tumor burden. The strategic approach after progression starts with a reassessment of tumor biological characteristics, ideally from hormonal receptors and the level of HER2 expression, to complete genomic profiling. Testing accuracy depends not only on educated and specialized staff but also on periodical internal and external quality control of testing and certification of pathological and molecular laboratories. These aspects are sometimes overridden; however, they are important prerequisites for reliable results [79].

The experts indicated preferred options for the identification of germline and somatic molecular aberrations before starting second-line treatment for the optimal treatment choice. Based on testing results, the efficacy and safety data from clinical trials should be balanced with toxicities and other potential negative aspects.

The Delphi exercise aimed to streamline the therapeutic decision. The translation of these recommendations in clinical practice may be limited at present by logistic barriers and uneven access to novel drugs and technologies, which remains a key problem both in Europe and the US [24,80,81,82,83].

In low-resource settings, we consider that better adoption of guidelines and structured approaches in daily practice, along with professional collaborations and partnerships, may help clinicians to adapt to these challenges [80]. Medical oncologists are required to remain flexible and use their clinical judgment to adapt to the mBC management according to local conditions, in addition to the variability of the patient and disease-related factors.

The medical community should remain vigilant regarding the quality of laboratory testing, as this is the crucial step in treatment selection. Moreover, treatment inertia should be overcome, and novel strategies should be adopted to improve survival and overall care in mBC.

## Figures and Tables

**Figure 1 cancers-17-01412-f001:**
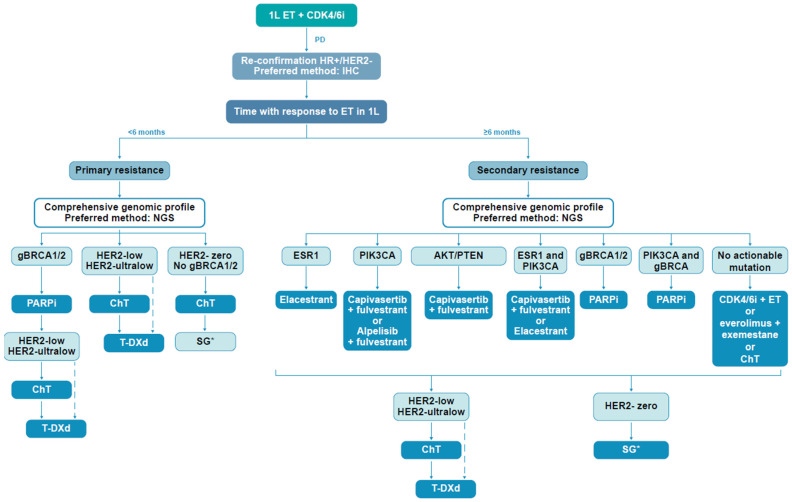
Overview of the 2L treatment recommendations in HR+/HER2− mBC based on mutations at disease progression. * SG in monotherapy should be considered for patients with HR+/HER2-0 mBC after at least two lines of ChT, based on the current indication approved in Europe, available at: https://www.ema.europa.eu/en/documents/product-information/trodelvy-epar-product-information_en.pdf (accessed on 8 April 2025). Dotted lines indicate the recommendation based on the DESTINY-Breast06 study [32]. At the time of manuscript development, this was not an approved indication for T-DXd. Abbreviations: 2L, second line; ChT, chemotherapy; mBC, metastatic breast cancer; SG, sacituzumab govitecan; T-DXd, trastuzumab deruxtecan.

**Table 1 cancers-17-01412-t001:** Summary of R1 statements, the initial level of agreement, relevant discussion during the kick-off (KO) meeting, and decisions for R2 statements.

R1 Statement	Level of Agreement Before KO Meeting	Summary of Suggestions and Comments	R2 Statement	Level of Agreement After KO Meeting
**Biomarker testing**				
**1. The NGS should be considered to guide 2L therapy selection for patients with HR+ HER2− mBC.**	**30% SA** **30% A** 20% D 10% N	NGS is the method of choice, if available and indicated for specific biomarkers Not all biomarkers can be identified with NGS After progression on 1L, biomarkers should be known in the treatment decision process	**1. (If available) NGS technique should be considered for biomarker testing.**	**100%**
			**2. Biomarkers should be known before the selection of 2L treatment.**	**100%**
**2. *BRCA1*/2 mutation testing should be performed at the initial diagnosis of metastatic disease to assess the potential benefit of PARP inhibitors and other targeted therapies in the 2L and subsequent treatment settings.**	**90% SA** **10% A**	Germline to be added Ideally, testing should be done at diagnosis (not at the diagnosis of metastatic disease) PARP-related text to be deleted	**3. Germline *BRCA1/2* mutation testing should be performed at the latest of the initial diagnosis or at metastatic disease, if not carried out before.**	**100%**
**3. *PIK3CA*, *AKT* mutation, and *PTEN* loss testing should be performed at the time of initial diagnosis of metastatic disease to determine eligibility for therapy in both 1L and 2L treatment settings.**	**90% SA** **10% A**		**4. *PIK3CA*, *AKT* mutation, and *PTEN* loss testing should be performed at the time of initial diagnosis of metastatic disease to determine eligibility for therapy in both 1L and 2L treatment settings.**	**100%**
**4. NGS is recommended due to its comprehensive coverage and high sensitivity for the detection of *AKT*, *PIK3CA*, *PTEN*, and *BRCA1/2* mutations in HR+ HER2− mBC.**	**60% SA** **40% A**	Listing of actionable mutations to be replaced with targeted mutations	**5. NGS is recommended due to its comprehensive coverage and high sensitivity for the detection of targeted mutations in HR+ HER2− mBC.**	**100%**
**5. Testing for *ESR1* mutations should be performed at the time of disease progression on 1L ET to guide the selection of 2L therapeutic options.**	**50% SA** **30% A** 20% D	*ESR1* mutation is dynamic, and testing should be repeated during the course of the disease to guide the treatment selection Delete 1L	**6. Testing for *ESR1* mutations should be performed at the time of disease progression on ET to guide the selection of further lines of therapeutic options.**	**100%**
**6. PCR is recommended for cost-effective testing of specific known mutations in biomarkers such as *PIK3CA*, where the target mutations are well characterized.**	**30% SA** **70% A**		**7. PCR is recommended for cost-effective testing of specific known mutations in biomarkers such as *PIK3CA*, where the target mutations are well characterized.**	**100%**
**7. Biomarker testing and related consultations should be available and reimbursed to encourage both patients and health care providers (HCPs) to utilize these essential services.**	**70% SA** **30% A**		**8. Biomarker testing and related consultations should be available and reimbursed to encourage both patients and HCPs to utilize these essential services.**	**100%**
**8. The primary barriers to the widespread adoption of genetic testing for HR+ mBC include technological challenges such as the complexity of testing procedures, lengthy turnaround times, and financial obstacles such as high costs and inconsistent insurance coverage.**	**40% SA** **50% A** 10% N	Quality control of genetic laboratories is a key aspect for the accuracy of testing. A new statement was added to reflect the need for standardization and certification of laboratories across the region	**9. The primary barriers to the widespread adoption of genetic testing for HR+ mBC include technological challenges such as the complexity of testing procedures and lengthy turnaround times, financial obstacles such as high costs and inconsistent insurance coverage.**	**90%**
			**10. Genetic testing with predictive value for HR+ mBC should be carried out in a certified laboratory with external and internal quality control.**	**100%**
**Selection of 2L treatment at PD on 1L ET + CDK4/6i, at ≥6 months after initiation of ET for HR+/HER2− mBC whilst on ET** **(secondary endocrine resistance)**				
**9. If confirmed *PIK3CA* mutations, capivasertib in combination with fulvestrant should be considered as a preferred 2L treatment option.**	**30% SA** **40% A** 10% D 20% N		**11. If *PIK3CA* mutations are confirmed, capivasertib in combination with fulvestrant should be considered as a preferred 2L treatment option.**	**70%**
**10. If confirmed *PIK3CA* mutations, alpelisib in combination with fulvestrant should be considered as a preferred 2L treatment option.**	**50% A** 20% D 20% SD 10% N	Rephrasing the statement was necessary. The context is that both drugs (capivsertib and alpelisib) are available	**12. In cases of confirmed *PIK3CA* mutation, capivasertib or alpelisib in combination with fulvestrant would be considered in the 2L treatment.**	**100%**
			**13. The use of capivasertib is the preferred 2L treatment over alpelisib due to the toxicity profile.**	**100%**
**11. If at risk for or have pre-existing diabetes, capivasertib should be preferred over alpelisib for 2L therapy due to the significant risk of hyperglycemia associated with alpelisib.**	**40% SA** **40% A** 20% N		**14. If at risk for or have pre-existing diabetes, capivasertib should be preferred over alpelisib for 2L therapy due to the significant risk of hyperglycemia associated with alpelisib.**	**100%**
**12. If both *ESR1* mutation and *PIK3CA* mutation, alpelisib should be considered as preferred 2L treatment option.**	**40% A** 10% D 20% SD 30% N	Not all three drugs are currently available in practice, and this might have been a challenge in reaching the level of consensus The agreement should be expressed considering an ideal situation, where all drugs are available and the clinician should choose one drug as the preferred option (in both ESR1 and PIK3CA mutations are present) S12, S13, and S14 will be sent to repeat the voting, in a different order	**15. If both *ESR1* mutation and *PIK3CA* mutation, elacestrant should be considered as preferred 2L treatment option.**	30% SA + A 40% D 30% N
**13. If both *ESR1* mutation and *PIK3CA* mutation, capivasertib should be considered as preferred 2L treatment option.**	**20% SA** **30% A** 20% D 30% N		**16. If both *ESR1* mutation and *PIK3CA* mutation, capivasertib should be considered as preferred 2L treatment option.**	**70%**
**14. If both *ESR1* mutation and *PIK3CA* mutation, elacestrant should be considered as preferred 2L treatment option.**	**20% SA** **20% A** 10% D 30% SD 20% N		**17. If both *ESR1* mutation and *PIK3CA* mutation, alpelisib should be considered as preferred 2L treatment option.**	**70%**
**15. If without *PIK3CA* and with alterations in the *AKT1* or *PTEN* pathway, capivasertib should be considered as a 2L treatment option.**	**60% SA** **20% A** 20% N		**18. If without *PIK3CA* and with alterations in the *AKT1* or *PTEN* pathway, capivasertib should be considered as a 2L treatment option.**	**80%**
**16. If with *ESR1* mutation, the use of elacestrant should be considered as the preferred 2L treatment option.**	**40% SA** **30% A** 10% D 20% N	Without *PIK3CA* mutation to be added	**19. If with *ESR1* mutation and without *PIK3CA* mutation, the use of elacestrant should be considered as the preferred 2L treatment option.**	**70%**
**17. If *gBRCA1/2* mutations, the use of olaparib or talazoparib should be considered as preferred 2L treatment.**	**50% SA** **30% A** 20% N		**20. If *gBRCA1*/*2* mutations, the use of olaparib or talazoparib should be considered as preferred 2L treatment.**	**80%**
**18. If both *gBRCA* mutations and *PIK3CA* pathway alterations, olaparib or talazoparib should be considered as the preferred 2L treatment option.**	**30% SA** **50% A** 20% N		**21. If both *gBRCA* mutations and *PIK3CA* pathway alterations, olaparib or talazoparib should be considered as the preferred 2L treatment option.**	**80%**
**19. If no actionable biomarkers are positive, everolimus combined with exemestane could be considered as a 2L option.**	**20% SA** **50% A** 30% N		**22. If no actionable biomarkers are positive, everolimus combined with exemestane could be considered as a 2L option.**	**70%**
**20. In patients with no actionable biomarkers, treatment rechallenge with an alternative CDK4/6i and change of endocrine partner should be considered after disease progression during 1L CDK4/6i + ET.**	**10% SA** **70% A** 20% N	“should” to be replaced with “could”, considering the results from the postMONARCH and MAINTAIN trials “rechallenge” to be deleted, following the discussion on S21	**23. In patients with no actionable biomarkers, treatment with an alternative CDK4/6i and change of endocrine partner could be considered after disease progression during 1L CDK4/6i + ET.**	**80%**
**21. In patients with actionable biomarkers, treatment rechallenge with an alternative CDK4/6i and change of endocrine partner should be considered after disease progression during 1L CDK4/6i + ET.**	**10% SA** **20% A** 40% D 10% SD 20% N	The statement was rephrased for more clarity: “rechallenge” to be deleted; “clear progression” to be added, to avoid misleading about oligoprogression	**24. In patients with actionable biomarkers and clear progression on CDK4/6i + ET, treatment with alternative CDK4/6i and change of endocrine partner should be considered.** * Only for this statement consensus was reached by disagreement.	**100% ***
**22. Fulvestrant monotherapy should be considered as a 2L treatment option, particularly in elderly and frail patients. This approach is suitable for those who may not tolerate more aggressive treatments or combination therapies due to their overall health status and comorbidities.**	**10% SA** **40% A** 20% SD 30% N	Suggestion for change: Fulvestrant monotherapy is suitable only for those patients not tolerating more aggressive treatment or combination therapy due to their health status and comorbidities (particularly elderly and frail patients)	**25. Fulvestrant monotherapy is appropriate only in those patients who cannot tolerate more aggressive treatment or combination therapy due to their health status and comorbidities (particularly elderly and frail patients).**	**100%**
**23. When considering the choice between alpelisib and capivasertib for 2L therapy in HR+ HER2− metastatic breast cancer, the toxicity profile and patient comorbidities should be carefully evaluated to optimize treatment tolerance and adherence.**	**50% SA** **30% A** 20% N		**26. When considering the choice between alpelisib and capivasertib for 2L therapy in HR+ HER2− mBC, the toxicity profile and patient comorbidities should be carefully evaluated to optimize treatment tolerance and adherence.**	**80%**
**24. Routine monitoring for hyperglycemia for patients on alpelisib and capivasertib is essential to identify elevated glucose levels early and initiate appropriate management to prevent complications such as CV events, infections, and impaired wound healing.**	**70% SA** **10% A** 20% N		**27. Routine monitoring for hyperglycemia for patients on alpelisib and capivasertib is essential to identify elevated glucose levels early and initiate appropriate management to prevent complications such as CV events, infections, and impaired wound healing.**	**80%**
**25. Non-sedating oral antihistamines should be considered for patients on alpelisib and capivasertib prophylactically.**	**40% SA** **30% A** 30% N		**28. Non-sedating oral antihistamines should be considered for patients on alpelisib and capivasertib prophylactically.**	**70%**
**26. Proactive management of diarrhea is crucial in patients on alpelisib and capivasertib, involving the early administration of anti-diarrheal agents, dietary modifications, and adequate hydration to prevent dehydration and electrolyte imbalances.**	**50% SA** **20% A** 30% N		**29. Proactive management of diarrhea is crucial in patients on alpelisib and capivasertib, involving the early administration of anti-diarrheal agents, dietary modifications, and adequate hydration to prevent dehydration and electrolyte imbalances.**	**70%**
**Selection of 2L treatment at PD on 1L ET + CDK4/6i, at <6 months after initiation of ET for HR+/HER2− mBC, whilst on ET (primary resistance in metastatic setting)**				
**27. Fast progression within 6 months of 1L therapy indicates a more aggressive disease biology, necessitating a swift shift to more aggressive or alternative treatment options, such as chemotherapy.**	**40% SA** **20% A** 20% D 20% N	Clarification of the statement, and then repeat the voting	**30. Fast progression within 6 months of 1L therapy indicates a more aggressive disease biology, necessitating a swift shift to an alternate systemic treatment.**	**80%**
**28. In cases of rapid progression on 1L CDK4/6i, *BRCA* testing should be carried out before 2L treatment decision.**	**40% SA** **40% A** 20% N	Germline to be added	**31. In cases of rapid progression on 1L CDK4/6i, g*BRCA* testing should be done before 2L treatment decision.**	**80%**
**29. If no actionable biomarkers are positive, chemotherapy should be considered as a 2L option.**	**20% SA** **60% A** 20% N	*Should* be replaced with *could* Or *should be considered in selected patients*	**32. If no actionable biomarkers are positive, chemotherapy could be considered as a 2L option.**	**80%**
**30. Single-agent taxanes should be considered in patients with significant tumor burden due to their proven efficacy and ability to provide rapid disease control.**	**20% SA** **30% A** 10% D 40% N	These statements will not be included in the final statement list		
**31. Single-agent taxane should be considered in patients who can tolerate intravenous administration and its associated side effects.**	**20% SA** **30% A** 10% D 40% N			
**32. Capecitabine should be considered in patients with slower disease progression and low tumor burden where immediate, aggressive treatment is not critical.**	**30% SA** **40% A** 30% N			
**33. Capecitabine should be considered in patients who prefer oral therapy and have previously been treated with anthracyclines and taxanes.**	**50% SA** **30% A** 20% N			
**34. Fast progressors on CDK4/6i should be considered for clinical trials, investigating novel therapies and combinations to provide access to potentially more effective treatments.**	**40% SA** **40% A** 20% N	Antibody drug conjugates (ADCs) should be emphasized as options for chemotherapy When available, depending on overall timelines, the results from the DESTINY-Breast06 study will be referenced in the text of the manuscript. However, the results may be used in the recommendation (statement) only after regulatory approval.	**33. Fast progressors on CDK4/6i should be considered for clinical trials, investigating novel therapies and combinations to provide access to potentially more effective treatments.**	**80%**
**Selection of post-2L treatment options (PD after minimum 2 lines of therapy for aBC)**				
**35. If HER2-low and no positive actionable biomarkers, trastuzumab deruxtecan (T-DXd) should be considered after one line of chemotherapy.**	**70% SA** 10% A 20% SA	After discussion for clarification, all experts agreed with the statement.	**34. If HER2 expression is low and no positive actionable biomarkers, trastuzumab deruxtecan (T-DXd) should be considered after one line of chemotherapy.**	**100%**
**36. If HER2-zero and no positive actionable biomarkers, sacituzumab govitecan (SG) should be considered after two lines of chemotherapy.**	**40% SA** **30% A** 20% SD 10% N		**35. If HER2-zero and no positive actionable biomarkers, sacituzumab govitecan (SG) should be considered after two lines of chemotherapy.**	**70%**
**37. In patients initially diagnosed with HR+ HER2− (IHC = 0) breast cancer, re-biopsy of metastasis, if possible, or re-testing of HER2 expression from the primary tumor should be considered at the time of progression.**	**70% SA** **30% A**	Re-testing or reassessment Not necessarily from the primary tumor, it can be from the tumoral tissue	**36. In patients initially diagnosed with HR+ HER2− (IHC = 0) breast cancer, re-biopsy of metastasis, if possible, or re-testing/re-assessing of HER2 expression should be considered at the time of progression.**	**100%**
**38. Changes in HER2 status can occur and may influence subsequent treatment decisions.**	**70% SA** **30% A**		**37. Changes in HER2 status can occur and may influence subsequent treatment decisions.**	**100%**
**39. T-DXd is effective in the treatment of HER2-low mBC, regardless of breast tumor sample type, used to determine HER2 status (primary tumor or metastasis).**	**50% SA** **30% A** 20% SD		**38. T-DXd is effective in the treatment of HER2-low mBC, regardless of breast tumor sample type, used to determine HER2 status (primary tumor or metastasis).**	**80%**

Answers from 10 experts represented the basis for the percentage calculation. Abbreviations: 1L, first line; 2L, second line; A, agree; ADC, antibody-drug conjugate; D, disagree; g, germline; HER2, human epidermal growth factor receptor 2; HR, hormone receptor; IHC, immunohistochemistry; KO, kick-off; mBC, metastatic breast cancer; N, neutral; NGS, new generation sequencing; PCR, polymerase chain reaction; PD, progression of the disease; R, round of delphi process; SA, strongly agree; SD, strongly disagree; SG, Sacituzumab govitecan; T-DXd, trastuzumab deruxtecan. * Only for this statement consensus was reached by disagreement.

## Data Availability

All data used in the manuscript are from already published scientific abstracts and articles.

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
