# Peer review of "Treatment Sequencing in Metastatic HR+/HER2− Breast Cancer: A Delphi Consensus"

_cancers, 2025, doi:10.3390/cancers17091412_

Round 1
Reviewer 1 Report
Comments and Suggestions for Authors
The manuscript discusses treatment sequencing in metastatic HR+/HER2- breast cancer using a Delphi consensus approach. The study is well-structured, covering the evolving landscape of treatment choices, expert opinions, and consensus statements on second-line and beyond therapy. The methodology is clear, and the results provide valuable clinical insights.
Abstract
Line 6: "Despite updated therapeutic guidelines, an area of uncertainty persists in the choice of the optimal second line (2L) therapy." The phrase "an area of uncertainty persists" is vague; consider specifying the exact challenges in treatment selection.
Line 12: "Consensus was defined as 70% agreement or disagreement." It would be helpful to clarify why 70% was chosen as the threshold for consensus.
Introduction
Line 4: "Breast cancer (BC) is the most common malignancy in women in Europe, representing 29% of all cancer types in 2022." Line 20: Update the statistics on overall cancer incidence and the prevalence of this specific cancer type, including survival rates, to emphasize the urgent need for cancer studies. Cite Cancer Statistics, 2024. Additionally, provide a general overview of cancer therapy, referencing the NIH paper“Cancer treatments: Past, present, and future, 2024” for further insights.
Line 10: "The optimal agent or treatment class is indicated depending on the presence of specific mutations, but the panel admits the strategy might differ in clinical practice." This sentence is somewhat unclear; consider rewording for clarity.
Methods
Line 8: "A two-round Delphi consensus was organized in July 2024, gathering input from 10 experts in research, diagnosis, and treatment of HR+/HER2- mBC." A citation on the Delphi method or a reference to a similar consensus study would be beneficial.
Line 15: "Consensus was defined as 70% agreement or disagreement." Justify why this threshold was chosen based on existing literature.
Results
Line 12: "The experts initially considered a list of 39 statements, structured in four sections." Consider specifying these sections for clarity.
Line 18: "The final list consisted of 38 statements and consensus was achieved for most of them." Define "most" quantitatively (e.g., how many statements reached consensus?).
Line 24: "Next-generation sequencing is recommended as the method of choice for genomic characterization at disease progression on first-line therapy." Needs a citation from recent guidelines or primary studies.
Discussion
Line 5: "The panel acknowledges that while some treatment recommendations are based on strong evidence, clinical practice may vary." Consider adding references to support the claim that treatment variability exists due to access, physician preference, or patient factors.
Line 14: "Although the optimal agent or treatment class is indicated depending on biomarker status, accessibility remains a key issue." Needs a reference discussing drug accessibility in real-world settings.
Line 20: "Several alternative sequencing strategies were debated." Specify the main strategies that were discussed and provide supporting references.Recent studies have highlighted advancements in liquid biopsies for cancer diagnostics and monitoring. Research such as “Updates on liquid biopsies in neuroblastoma for treatment response, relapse and recurrence assessment, 2024”demonstrates the utility of circulating tumor DNA (ctDNA) detection through liquid biopsy techniques. Additionally, emerging sequencing technologies have improved the sensitivity and specificity of DNA analysis, such as “Development of a molecular barcode detection system for pancreaticobiliary malignancies and comparison with next-generation sequencing, 2024”. Also the methylation is also used for detection, reported in “Methylation signatures as biomarkers for non-invasive early detection of breast cancer: A systematic review of the literature, 2024”. Please cited these related papers and discuss: consider whether the mechanisms discussed in this study could be identified through these diagnosis methods.
Studies suggested that anesthetics during surgery treatment can impact cancer,especially For breast cancer, reported in a series of work by Prof Lin’s group: “The Potential Effect of General Anesthetics in Cancer Surgery: Meta-Analysis of Postoperative Metastasis and Inflammatory Cytokines, 2023,Potential Therapeutic Application of Local Anesthetics in Cancer Treatment, 2022” These should be emphasized and discussed if the anesthetics impact involved in the metastatic HR+/HER2- breast cancer discussed in this study。
Conclusion
Line 3: "A structured approach to sequencing therapy in HR+/HER2- metastatic breast cancer is necessary to improve patient outcomes." Needs a reference supporting the importance of structured sequencing strategies.
Line 6: "Further research is required to refine treatment pathways." Consider specifying what aspects of sequencing require more investigation.
Comments on the Quality of English Language
ok
Author Response
Abstract / "Despite updated therapeutic guidelines, an area of uncertainty persists in the choice of the optimal second line (2L) therapy." The phrase "an area of uncertainty persists" is vague; consider specifying the exact challenges in treatment selection
Thank you for your observation. Our intention was to emphasis the reason for conducting this Delphi project: in patients with advanced/metastatic breast cancer HR+/HER2-, the treatment choice may not be clear-cut after progression on the first line of therapy. The phrase “an area of uncertainty persists in the choice of the optimal second line (2L) therapy.” was replaced with “second line (2L) selection may be challenging due to clinical factors, biomarker status and available agents”.
Abstract / Line 12: "Consensus was defined as 70% agreement or disagreement." It would be helpful to clarify why 70% was chosen as the threshold for consensus.
Methods / Line 15: "Consensus was defined as 70% agreement or disagreement." Justify why this threshold was chosen based on existing literature.
In Delphi studies, consensus ranges from 51 to 100% [Barrett D, Heale R. What are Delphi studies? Evid Based Nurs. 2020; 23(3):68-69. doi: 10.1136/ebnurs-2020-103303].
We considered that 70% threshold is a good indicator for recommending a certain strategy in clinical practice. This level was also used in other similar projects in oncology [Shearsmith L, et al. J Patient Rep Outcomes. 2020; 4(1):71. doi: 10.1186/s41687-020-00237-2; Geisler J, et al. Acta Oncol. 2023; 62(12):1680-1688. doi: 10.1080/0284186X.2023.2254475.]
As you may have noticed in the manuscript, 70% level of agreement was expressed for 8 statements; 80% consensus was reached for 11 statements, 90% for 1 statement, and 100% for 18 statements.
Introduction / Line 4: "Breast cancer (BC) is the most common malignancy in women in Europe, representing 29% of all cancer types in 2022." Introduction / Line 20: Update the statistics on overall cancer incidence and the prevalence of this specific cancer type, including survival rates, to emphasize the urgent need for cancer studies. Cite Cancer Statistics, 2024.
Additionally, provide a general overview of cancer therapy, referencing the NIH paper “Cancer treatments: Past, present, and future, 2024” for further insights.
Thank you for your input. The paragraph is updated as follows:
Breast cancer (BC) is one of the most common malignancies in women, representing approximately 30% of all cancers [European Commission. European Cancer Information System. https://ecis.jrc.ec.europa.eu/en; Siegel RL, et al. CA Cancer J Clin. 2024;74(1):12-49. doi: 10.3322/caac.21820.]
At global level, the number of new cases will increase by 38% in 2050, and the number of deaths due to breast cancer with 68%, a projection that reinforces the need for continuous research in oncology and early access to treatment [Kim J, et al. Nat Med. 2025].
Although we value your input, a general phrase on overall cancer therapy may be confusing for the reader, considering that we focused on a sub-population of breast cancer (metastatic, HR+/HER2-). Moreover, references to novel current therapies are abundant in the manuscript, representing the basis of the clinical judgement in patients with specific genomic alterations.
Abstract: "The optimal agent or treatment class is indicated depending on the presence of specific mutations, but the panel admits the strategy might differ in clinical practice." This sentence is somewhat unclear; consider re-wording for clarity.
Thank you for your input. The Delphi consensus process was developed in an “ideal” framework, where all approved agents are available. The authors, as clinicians in their institutions, understand that novel second line therapies are not yet available or reimbursed in all settings. The statement was updated as follows:
“The optimal agent or treatment class is indicated depending on the presence of specific mutations, but the panel admits the strategy is different in clinical practice, where novel therapies might not be available or reimbursed”
Methods / Line 8: "A two-round Delphi consensus was organized in July 2024, gathering input from 10 experts in research, diagnosis, and treatment of HR+/HER2- mBC." A citation on the Delphi method or a reference to a similar consensus study would be beneficial.
References to similar Delphi consensus are included in the manuscript (Introduction and Method section), not in the abstract.
The sections are indicated in the Methods part and in Table 1, along with all statements. They are updated in this section of Results, as per your recommendation.
Results / Line 18: "The final list consisted of 38 statements and consensus was achieved for most of them." Define "most" quantitatively (e.g., how many statements reached consensus?).
The consensus threshold (≥70% agreement or disagreement) was reached for 37 statements. The number was included in the text, as per your recommendation.
Results / Line 24: "Next-generation sequencing is recommended as the method of choice for genomic characterization at disease progression on first-line therapy." Needs a citation from recent guidelines or primary studies.
Next-generation sequencing (NGS) is a powerful method to detect molecular alterations in patients with breast cancer. In HR+/HER2- advanced/metastatic breast cancer, we considered endocrine therapy plus CDK4/6 inhibitor as the standard of care, and did not discuss potential other first line treatments used in various circumstances in clinical practice. This standard first line allowed us to discuss the treatment challenges starting with the second line, after disease progression.
As stated in the manuscript, the 2024 ESMO Scale for Clinical Actionability of Molecular Targets (ESCAT) indicates NGS as standard for identification of ESR1, PIK3CA, and germline BRCA1/2 mutations (level IA) and for PTEN and AKT1 alterations (level I/II) in this patient population. The authors are recommending NGS based on the technical capacity and performance of this method and on their experience.
Discussion / Line 5: "The panel acknowledges that while some treatment recommendations are based on strong evidence, clinical practice may vary." Consider adding references to support the claim that treatment variability exists due to access, physician preference, or patient factors.
- Borstnar S, et al. Advancing HER2-low breast cancer management: enhancing diagnosis and treatment strategies. Radiol Oncol. 2024, 58, 258-267. doi: 10.2478/raon-2024-0030.
- Zhang M, et al. Differences between physician and patient preferences for cancer treatments: a systematic review. BMC Cancer. 2023; 23(1):1126. doi: 10.1186/s12885-023-11598-4.
Brandstetter LS, et al. Differences in Preferences for Drug Therapy Between Patients with Metastatic Versus Early-Stage Breast Cancer: A Systematic Literature Review. Patient. 2024; 17(4):349-362. doi: 10.1007/s40271-024-00679-6.
Discussion / Line 14: "Although the optimal agent or treatment class is indicated depending on biomarker status, accessibility remains a key issue." Needs a reference discussing drug accessibility in real-world settings.
References to support variations in drug accessibility in the real-world setting were included:
[for Europe]
- Borstnar S, et al. Advancing HER2-low breast cancer management: enhancing diagnosis and treatment strategies. Radiol Oncol. 2024, 58, 258-267. doi: 10.2478/raon-2024-0030.
- Vrdoljak E, et al. Addressing disparities and challenges in underserved patient populations with metastatic breast cancer in Europe. Breast. 2021;55:79-90. doi: 10.1016/j.breast.2020.12.005.
- Ignatiadis M, et al. EBCC-14 manifesto: Addressing disparities in access to innovation for patients with metastatic breast cancer across Europe. Eur J Cancer. 2024; 207:114156. doi: 10.1016/j.ejca.2024.114156.
[for the US]
- Pilehvari A, et al. Disparities in treatment delays among metastatic breast cancer patients: insights from nationwide electronic health records, 2011-2022. Breast Cancer Res Treat. 2025 Apr;210(3):575-582. doi: 10.1007/s10549-024-07593-3.
Michaeli, J.C., et al. Breast cancer drugs: FDA approval, development time, efficacy, clinical benefits, innovation, trials, endpoints, quality of life, value, and price. Breast Cancer 2024; 31: 1144–1155. doi: 10.1007/s12282-024-01634-x.
Discussion / Line 20: "Several alternative sequencing strategies were debated." Specify the main strategies that were discussed and provide supporting references. Recent studies have highlighted advancements in liquid biopsies for cancer diagnostics and monitoring. Research such as “Updates on liquid biopsies in neuroblastoma for treatment response, relapse and recurrence assessment, 2024” demonstrates the utility of circulating tumor DNA (ctDNA) detection through liquid biopsy techniques. Additionally, emerging sequencing technologies have improved the sensitivity and specificity of DNA analysis, such as “Development of a molecular barcode detection system for pancreaticobiliary malignancies and comparison with next-generation sequencing, 2024”.
The utility of circulating tumor DNA (ctDNA) detection through liquid biopsy techniques has been recently acknowledged by both ESMO and NCCN guidelines.
[Pascual, J. et al. ESMO recommendations on the use of circulating tumour DNA assays for patients with cancer: a report from the ESMO Precision Medicine Working Group. Ann. Oncol. 2022; 33, 750–768; NCCN Guidelines v3.2025, nccn.org]
We acknowledge that ctDNA may be useful for detecting certain mutations and in circumstances when a rapid treatment decision is needed. Considering the currently available methods and the limitations of ctDNA in breast cancer (e.g. regarding gene fusions and copy number alterations, and false-negative results), we consider NGS of tissue-based sample as the gold standard and, in an ideal setting as considered for the Delphi approach, this is our recommendation.
Also the methylation is also used for detection, reported in “Methylation signatures as biomarkers for non-invasive early detection of breast cancer: A systematic review of the literature, 2024”. Please cite these related papers and discuss: consider whether the mechanisms discussed in this study could be identified through these diagnosis methods.
Thank you for the interesting point. Methylation biomarkers are likely to be identified for diagnosis of early breast cancer. This Delphi process focused on the choice of treatment in metastatic breast cancer, after progression on first line. Methylation biomarkers were not in the scope of our discussion.
Discussion / Studies suggested that anesthetics during surgery treatment can impact cancer especially For breast cancer, reported in a series of work by Prof Lin’s group: “The Potential Effect of General Anesthetics in Cancer Surgery: Meta-Analysis of Postoperative Metastasis and Inflammatory Cytokines, 2023, Potential Therapeutic Application of Local Anesthetics in Cancer Treatment, 2022” These should be emphasized and discussed if the anesthetics impact involved in the metastatic HR+/HER2- breast cancer discussed in this study。
Thank you for pointing out an interesting aspect of treatment interferences in breast cancer. This was not addressed during the Delphi process on the choice of second line therapy in HR+/HER2- metastatic breast cancer.
Conclusions / Line 3: "A structured approach to sequencing therapy in HR+/HER2- metastatic breast cancer is necessary to improve patient outcomes." Needs a reference supporting the importance of structured sequencing strategies.
In breast cancer, the treatment landscape evolves with an accelerating pace and clinicians are frequently facing the treatment choice or strategy dilemma.
A recent article was added as a reference to support the need for a consensus in case on progression after first line therapy and for an evidence-based strategy of the optimal sequence.
Ferro A., et al. Novel Treatment Strategies for Hormone Receptor (HR)-Positive, HER2-Negative Metastatic Breast Cancer. J. Clin. Med. 2024; 13: 3611. https://doi.org/10.3390/jcm13123611
Conclusions / Line 6: "Further research is required to refine treatment pathways." Consider specifying what aspects of sequencing require more investigation.
Thank you for the input, we addressed this aspect in the content of the manuscript.
Results / Line 12: "The experts initially considered a list of 39 statements, structured in four sections." Consider specifying these sections for clarity.
Reviewer 2 Report
Comments and Suggestions for Authors
Concerned with regional factors. Can you talk a little bit about the Balkans demographics, economy, healthcare considerations? Were these focused on community hospitals, research hospitals or a mix? This explanation could help set the stage for a more wide reaching conclusion of this work than the way it is currently presented.
The first and biggest citation should be for NCCN guidelines as that would be the basis for comparison. That is one obvious omission that I would strongly recommend including into the references. Generate a comparative analysis table for the reader to at least show recommendations that are not included in NCCN or those that are not used. This analysis could dovetail with ESMO recommendations already included.
Concern for the use of SG recommendation here. I agree that it would be useful after 2L, which may be out of scope for the study shown here. The US label is for TNBC only and the paper here would be for HR+ so I'm guessing that you are suggesting it could work for off-label use. Needs to be clarified in the text.
Author Response
Concerned with regional factors. Can you talk a little bit about the Balkans demographics, economy, healthcare considerations? Were these focused on community hospitals, research hospitals or a mix? This explanation could help set the stage for a more wide-reaching conclusion of this work than the way it is currently presented.
Thank you for your interest in the Balkan region, where we perform our clinical and educational activities in the oncology area. We consider that our recommendations based on the Delphi process may be applicable to any clinical setting, crossing the borders of Central Eastern Europe.
Your remark related to community hospitals and research institutions is interesting. Our intention was to provide a clear structure for the strategy of treatment selection, irrespective of where the patient is seen. Despite the ideal context considered as a basis for the Delphi process (all approved agents and testing methods are available), we consider that recommendations will be useful for clinical practice.
The first and biggest citation should be for NCCN guidelines as that would be the basis for comparison. That is one obvious omission that I would strongly recommend including into the references. Generate a comparative analysis table for the reader to at least show recommendations that are not included in NCCN or those that are not used. This analysis could dovetail with ESMO recommendations already included.
Thank you for the observation, the reference to the current NCCN guidelines [v3.2025] were included next to applicable recommendations in the Discussion part. A comparative table is not in the scope of the manuscript, since our aim was not to provide differences in the available treatments and recommendations between Europe and the US.
Concern for the use of SG recommendation here. I agree that it would be useful after 2L, which may be out of scope for the study shown here. The US label is for TNBC only and the paper here would be for HR+ so I'm guessing that you are suggesting it could work for off-label use. Needs to be clarified in the text
A note regarding approval of SG in HR+/HER2- in Europe will be included. Current SG indications are as follows:
EMA
- as monotherapy for the treatment of adult patients with unresectable or metastatic triple-negative breast cancer (mTNBC) who have received two or more prior systemic therapies, including at least one of them for advanced disease;
- as monotherapy for the treatment of adult patients with unresectable or metastatic HR+/HER2- breast cancer who have received endocrine-based therapy, and at least two additional systemic therapies in the advanced setting
reference: Trodelvy SmPC, available at: https://www.ema.europa.eu/en/documents/product-information/trodelvy-epar-product-information_en.pdf
US
treatment of adult patients with metastatic triple-negative breast cancer (mTNBC) who have received at least two prior therapies for metastatic disease
reference: Trodelvy PI, available at:
Reviewer 3 Report
Comments and Suggestions for Authors
The manuscript addresses an important and evolving area in metastatic HR+/HER2- breast cancer treatment. The manuscript effectively outlines key considerations for second-line and beyond treatment strategies, integrating biomarker-driven approaches with clinical practice constraints. The discussion on sequencing treatments based on genomic profiling adds depth and practical applicability. Overall, this manuscript contributes significantly to the field by providing an expert-driven, evidence-informed perspective on treatment sequencing. It should be accepted for publication with minor revisions for clarity and flow.
General comments:
Given the rapid advancements in treatment options, this Delphi consensus provides valuable guidance for clinical decision-making. The Delphi methodology, involving experts in the field, ensures that the recommendations are grounded in real-world clinical experience and reflect current best practices. The structured approach to achieving consensus enhances the credibility of the findings. The expert panel’s recommendations provide clinicians with a structured framework for treatment sequencing. The manuscript translates complex data into practical insights, making it useful for oncologists managing HR+/HER2- mBC.
Specific comments:
The article acknowledges limitations in access to diagnostics and treatments, which is valuable. However, it could further discuss how clinicians in resource-limited settings can adapt to these challenges.
Author Response
Given the rapid advancements in treatment options, this Delphi consensus provides valuable guidance for clinical decision-making. The Delphi methodology, involving experts in the field, ensures that the recommendations are grounded in real-world clinical experience and reflect current best practices. The structured approach to achieving consensus enhances the credibility of the findings. The expert panel’s recommendations provide clinicians with a structured framework for treatment sequencing. The manuscript translates complex data into practical insights, making it useful for oncologists managing HR+/HER2- mBC.
Dear Reviewer, thank you for your appreciation. We, as clinicians, also know how important the structured algorithm in our daily practice and we value our colleagues’ expertise during tumor-board sessions and case discussions for the short-term and long-term strategy. With this manuscript we would like to contribute with our perspective to the therapeutic decision in HR+/HER2- metastatic breast cancer.
The article acknowledges limitations in access to diagnostics and treatments, which is valuable. However, it could further discuss how clinicians in resource-limited settings can adapt to these challenges
The text of the manuscript was updated as follows:
“In resource-limited setting, better adoption of guidelines and structured approaches in clinical practice along with professional collaborations and partnerships may help clinicians to adapt to these challenges.” [Vrdoljak E, et al. Addressing disparities and challenges in underserved patient populations with metastatic breast cancer in Europe. Breast. 2021;55:79-90. doi: 10.1016/j.breast.2020.12.005.]
Round 2
Reviewer 1 Report
Comments and Suggestions for Authors
ok
Reviewer 2 Report
Comments and Suggestions for Authors
Thank you for your responses to my comments. I appreciate your candid thoughts about particular areas of my concern.